# A systematic review of exergame usability as home-based balance training tool for older adults usability of exergames as home-based balance training

Candice Simões Pimenta de Medeiros[1]*, Luanna Bárbara Araújo Farias[1], Maria Clara do Lago Santana[1], Thaiana Barbosa Ferreira Pacheco[1], Rummenigge Rudson Dantas[2], Fabrícia Azevedo da Costa Cavalcanti[1]

1 Physical Therapy Department, Federal University of Rio Grande do Norte, Natal, Rio Grande do Norte, Brazil, 2 Digital Metropolis Institute, Federal University of Rio Grande do Norte, Natal, Rio Grande do Norte, Brazil

* candice_spmedeiros@yahoo.com.br

## Abstract

### Background

Exergames are a fun, viable, attractive, and safe way to engage in physical exercise for most patient populations, including older adults. Their use in the home environment enables an expanded understanding about its applicability and its impact on clinical outcomes that can contribute to improved functionality and quality of life in this population. This systematic review aimed to synthesize the evidence on the usability of exergames as a tool for home-based balance training in older adults.

### Methods

The search was realized in 6 electronic databases and were included 1) randomized controlled trials with exergames home-based training as intervention, 2) studies involving older adults (aged 60 years or older) described as having impaired static or dynamic balance, 3) that compared the effects of exergames to usual care, health education or no intervention, and 4) reported usability and balance outcomes. The Cochrane Risk of Bias tool for randomized trials version 2 and the Grading of Recommendations Assessment, Development, and Evaluation were used to evaluate the methodological quality of studies and levels of evidence for outcomes.

### Results

After screening 1107 records, we identified 4 trials were included. The usability score of exergames was classified as an acceptable, good, and feasible tool. The pooled effect indicated improvements in favor of the exergame group for functional balance by TUG test (MD = -5.90; 95%CI = -10.29 to -1.51) with low-certainty evidence and Tinetti scale (MD = 4.80; 95%CI = 3.36 to 6.24) with very low-certainty evidence. Analyzing the different immersion

**Data Availability Statement:** All relevant data are within the manuscript and its Supporting Information files.

**Funding:** The present work was carried out with the scholarship support of the Coordination for the Improvement of Higher Education Personnel – Brazil (CAPES)(Finance Code 001).

**Competing interests:** The authors have declared that no competing interests exist.

level, it was observed a significant difference in the experimental group for the immersive exergames (MD = -9.14; 95%CI = -15.51 to -2.77) with very low-certainty evidence.

## Conclusion

Exergames applied at home showed good usability and had significant effects on functional balance compared to usual care or no intervention, especially in the immersive modality.

## Trial registration

**PROSPERO registration number**: CRD42022343290.

## Introduction

The aging process is universal and has a significant impact on motor skills due to the progressive deterioration of sensory, cognitive, and motor functions, which affects balance, functionality, and mobility [1, 2]. Deficits in postural control have serious consequences for physical functioning, in addition to being significant predictors of falls in older adults [3, 4]. Postural control involves a complex and dynamic skill resulting from the interaction of sensory, perceptual, and motor processes, promoting postural orientation and balance [5, 6]. Functional balance represents the older adults' ability to maintain stability and balance during daily activities, both statically or dynamically, impacting their independence and quality of life [7, 8].

An effective method for enhancing gait and balance is through the utilization of virtual reality (VR)-based exercises, commonly referred to as exergames. Some studies have reported that exergames are a fun, viable, attractive, and safe way to engage in physical exercise in diverse populations [9–11], including older adults and in the context of fall prevention [12, 13]. Exergames enable the practice of physical exercise through interactions with motion sensors in a virtual environment [9], leading to increased levels of physical activity, fun, interaction, and motivation of players [14]. Adcock et al. [15] have elucidated that exergame training can be performed in various environments, including at home. Moreover, recent research has shown that home-based exergame training is widely accepted among older adults [15–18]. The home environment, with its comfort, security, and privacy, is where people express their personal identity and autonomy, impacting the physical activity levels of older adults [19, 20]. Thus, home-based exercise is vital for reducing fall risk, boosting aerobic fitness, and combatting sedentary behavior, thereby improving functional capacity and autonomy.

The International Organization for Standardization (ISO) 9241–11 defines usability based on measures of efficiency (resources required for effectiveness), effectiveness (accuracy and completeness), and satisfaction (comfort and acceptability) of a user when interacting with a tool in a specific context or environment [21, 22]. Understanding the usability of exergames in the home environment enables an expanded understanding of the applicability, dosage, game characteristics, and modalities of this therapeutic tool; as well as its impact on clinical outcomes that can contribute to improved functionality and quality of life in this population, reducing the risk of falls. Some studies highlight the need for more robust recommendations for the use of virtual reality at home, in residential facilities, and in long-term care institutions [23–25].

Considering the rapid growth of the older adult population and the high risk of fall episodes, improving the health status and independence of these individuals is extremely important [15], as well as expanding the understanding of the potential benefits of exergames in the

home environment for this population. No study in the literature has addressed the usability of exergame systems applied in the home context of older adults as a tool for balance training, considering the intervention characteristics and its direct impact on balance, mobility, quality of life, and adverse effects. Thus, this systematic review aimed to synthesize the evidence on the usability of exergames as a tool for home-based balance training in older adults.

## Materials and methods

This systematic review was registered on the PROSPERO database (CRD42022343290) and followed in accordance with Preferred Reporting Items for Systematic Reviews and Meta-Analysis (PRISMA) guidelines [26]. The systematic review protocol has been published and provides a full outline of the methods [27]. A summary of the methods is reported in this paper.

### Data sources

A database search was conducted from inception to December 2022 of MEDLINE (Pubmed), Web of Science, Embase (Elsevier), Scopus (Elsevier), ScienceDirect, and the Cochrane Central Register of Controlled Trials (CENTRAL). We will also conduct a search on ClinicalTrials.gov, the WHO International Clinical Trials Registry Platform (ICTRP), and ReBEC for ongoing or unpublished trials.

### Eligibility criteria

Studies included in this systematic review met the following criteria: (1) Randomized controlled trials (RCTs); (2) Studies that compare home-based balance training realized using exergames in older adults with health education interventions, usual care, or no intervention; (3) Studies on older adults (aged 60 years or older), who are described as having impaired static or dynamic balance using any subjective or objective assessment criteria (e.g. Berg Balance Scale, Timed Up and Go, Tinetti scale, force plate center of pressure, among others); (5) Studies conducted on older adults without associated neurological, orthopedic, cardiac, or rheumatic pathologies.

The home-based environment was characterized as: the home of the older adult, and housing environments such as senior citizens' clubs, elderly homes or retirement homes, residential care facilities, assisted living communities, and independent living centers will be considered. As these definitions might have different meanings for different individuals and places, we established that the older adults included in the study should exhibit functional independence and autonomy. Therefore, institutionalized, hospitalized, and nursing-home older adults, identified by the presence of significant functional dependence and/or bed restriction, were excluded from this study [27].

### Outcomes measures

The primary outcomes were postural balance and usability. This may include assessments with: (1) postural balance using instruments that analyze functional balance (the Berg Balance Scale, Timed Up and Go, functional reach test, force platform measures, Tinetti test, balance master system, among others)–these outcomes measure many different resources for postural control–and (2) usability, (e.g. system usability scale or any kind of questionnaire, scale, or report that describes the level of usability and adherence to exergame therapies). The secondary outcomes were safety (self-reported impression), mobility, quality of life, motivation, falls, and adverse events.

## Study selection

The screening for eligible studies was conducted independently by two reviewers (CSPM and LBAF). An electronic screening form was used, and screening occurred in stages: first, titles were screened, followed by abstracts and finally full-text articles were screened. Conflicts were resolved by consensus from CSPM, LBAF and MCLS. The studies were imported, managed, and filtered using the RAYYAN online database (RAYYAN Intelligent Systematic Review tool) [28].

## Data extraction

Data extraction was completed by two authors independently (CSPM and TBFP), and conflicts were resolved by a third author (MCLS). A data extraction form was developed to collect study characteristics and outcome data through discussions among all authors, and according to the PRISMA statement [26, 29]. When data were missing, study authors were contacted by email to provide further information.

The extracted data were transferred by one reviewer (TBFP) to the Review Manager 5.4.1 (RevMan) [30], recording the study characteristics: (1) Study information: year, author information, funding or sponsorship information, study type, journal name, study duration, study location, population, intervention, control, and outcome (PICO elements); (2) Methods: the study design, study setting, sample, randomization method, participant recruitment methods, allocation method, inclusion and exclusion criteria, and risk of bias; (3) Participant detail: descriptive characteristics including age, gender, race, and comorbidities; (4) Intervention: intervention type, exergame characteristics, immersion level, and game information; (5) Outcomes: outcomes specified and collected, primary and secondary outcomes and adverse events. Primary and secondary outcome data were extracted before intervention and post intervention time points.

## Assessment of risk of bias

The version 2 of the Cochrane Risk of Bias tool for randomized trials (RoB 2) was used to assess the risk of bias [31, 32]. The risk of bias was undertaken by two independent reviewers (CSPM and RRD) with conflicts resolved by a third reviewer (FACC). The risk of bias was classified as "high" or "low", or be labeled "some concerns" based on randomization process, deviations from intended interventions, missing outcome data, measurement of the outcome, and selection of the reported result [31, 32].

## Assessment of quality of evidence

The quality of evidence was assessed by two review authors (TBFP and FACC) using The Grading of Recommendations, Assessment, Development, and Evaluation (GRADE) [33]. The GRADE approach uses five domains–risk of bias, consistency of effect, imprecision, indirectness, and publication bias–, and four levels of certainty: high, moderate, low, and very low.

## Statistical analysis

We presented a narrative summary of the study results. The meta-analysis was conducted with Review Manager version 5.4 (Cochrane Collaboration, Oxford, UK) [30]. The Cochran $Chi^2$ test and $I^2$ statistic were used to assess the degree of heterogeneity. With $p < 0.05$ for the $Chi^2$ test and benchmarks for interpreting $I^2$: (1) unimportant: 0–40%; (2) moderate: 30–60%; (3) substantial: 50–90%; and (4) considerable: 75–100% [27, 32]. We used a random effects model for primary outcome. All outcomes were continuous, and the results were presented as mean

difference and 95% confidence intervals. Sensitivity and subgroup analyses were conducted to explore the sources of heterogeneity. Sensitivity analysis was conducted to observe changes by removing a single study.

# Results

## Flow of studies included in this review

The initial search of the databases resulted in 1107 studies (Fig 1). After removing duplicate papers, 622 studies were screened to analysis of title and abstract. 594 of the studies did not meet the inclusion criteria, and all of 28 articles to read in the full texts were in English. The analysis of the complete texts led to the exclusion of an additional 24 studies that did not meet the inclusion criteria: 8 were RCT protocols or other type of study; 6 did not assess the usability

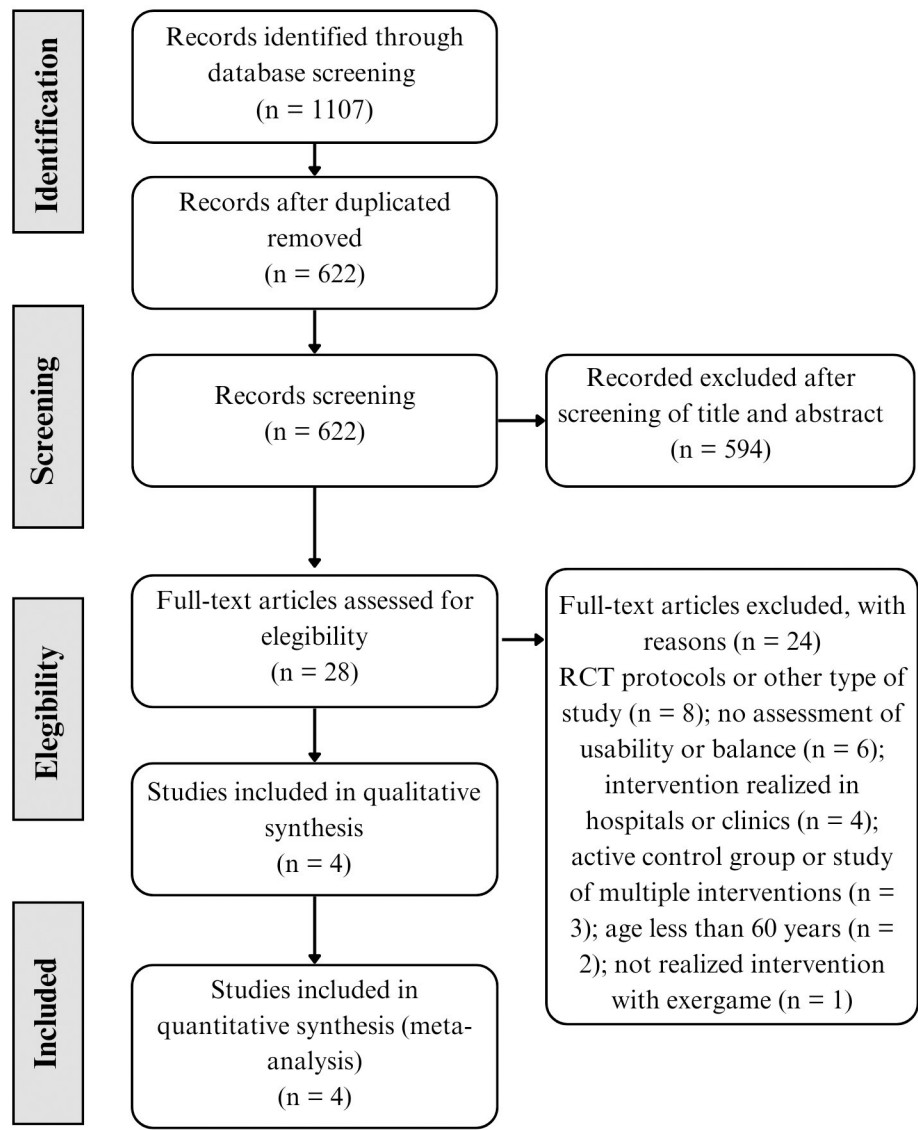

**Fig 1. Preferred reporting items for Systematic Reviews and Meta-Analyses flow of studies through the review.**

or balance; 4 were interventions realized in hospitals or clinics; 3 had an active control group or were study of multiple interventions; 2 had the study population aged less than 60 years; and 1 did not realize intervention with exergame. Hence, we incorporated the remaining 4 studies into the systematic review.

## Characteristics of participants and interventions

Studies included samples ranging from 12 to 136 participants, with a total of 202 older adults. Both males and females were included in the studies, all had a higher percentage of female participants [34–36]. One study included only female participants [37]. The exergame groups had a mean age of 83.93 ± 6 years, while the control groups (receiving usual care or no intervention) had an average age of 83.81 ± 5.8 years. Two studies were executed in a senior center [34, 37], one was in the home of the older adult [36], and one study was conducted in a senior living community [35].

Publication dates ranged from 2014 to 2022. Studies were conducted in five different countries: a single study was conducted in the USA [35]; two studies were conducted in Spain [34, 37], and one study was conducted across three countries: Germany, Spain and Australia [36].

Regarding the immersion level, two studies [35, 36] were conducted with semi-immersive virtual reality and two were immersive virtual reality (IVR) [34, 37]. Two studies used a HTC VIVE Pro[TM] Virtual Reality Headset an IVR device through the commercial game Box VR [34, 37], one study used the Microsoft Kinect to perform a serious game, the iStoppFall system, with the Bumble Bee Park, Hills & Skills, and Balance Bistro games [36]; and one study performed an interactive game-based virtual interface with wearable inertial sensors whose player is challenged to cross virtual obstacles appearing on the screen [35].

Interventions were carried out for 4 weeks [35], 10 weeks [34, 37] and 16 weeks [36]. The frequency of 3 times a week was observed in three studies [34, 36, 37] and one study [35] performed twice a week. Regarding training instructions, assistance, and supervision, in the study by Campo-Prieto, Cancela-Carral, and Rodríguez-Fuentes [34], the IVR equipment was installed three weeks prior to the intervention, users received training and instructions on handling, and the sessions were supervised. Campo-Prieto et al. [37] organized a meeting to facilitate an introduction to the IVR, providing instructions, explanations, and tranining practice on handling. In the study by Gschwind et al. [36], a trained research staff installed the iStopp-Falls system in the homes of older adults. Participants were instructed on correct and safe system use, phone support was available throughout intervention, and additional home visits were offered if required. Schwenk et al. [35] reported that the equipment was installed by the researchers, the balance tasks were explained to the participants during the first session, and the sessions were supervised.

The cumulative exergame exposure duration, calculated as the product of the number of sessions and the duration of each session, varied between 180 minutes [34, 37] and 2880 minutes [36], with an average duration of 270 minutes. In one study, the control group had no intervention [35] and in three studies, the control groups received usual care [34, 36, 37]. Details of study characteristics are summarized in Table 1.

## Outcomes

All studies measured balance with varied methods. Four studies used the Timed Up and Go (TUG) [34–37], two used Tinetti balance test [34, 37], and one used bipedal, semi-tandem, near-tandem and tandem stance [36]. One study used a force platform BalanSens™ to assess the center of pressure (CoP) in the conditions of eyes open and closed, during 30 seconds and

**Table 1. Summary of the characteristics of the included studies.**

| Study author; Country | Study characteristics | Intervention | Control | Dosage: duration; session length; frequency; volume of therapy | Outcomes measure |
|---|---|---|---|---|---|
| Campo-Prieto, Cancela-Carral and Rodríguez-Fuentes [34]; Spain | Sample: n = 24 participants (IVR group = 13; control group = 11) Mean age: IVR group = 85.05 ± 8.45; control group = 84.82 ± 8.1 Gender: IVR group = 84.61% female; control group = 90.90% female | Immersive Virtual Reality using HTC VIVE Pro™ commercial entertainment device using a Box VR game (commercial game) | Usual Care | 10 weeks; 6 minutes; 3x week; 180 minutes | Tinetti test Timed Up and Go test Five times sit-to-stand test Handgrip strength 12-Item Short Form Survey Simulator Sickness Questionnaire System Usability Scale Game Experience Questionnaire Satisfaction questionnaire |
| Campo-Prieto et. [37]; Spain | Sample: n = 12 participants (IVR group = 6; control group = 6) Mean age: IVR group = 91.67 ± 1.63; control group = 90.83 ± 2.64 Gender: IVR group and control group = 100% female | Immersive Virtual Reality using HTC VIVE Pro™ commercial entertainment device using a Box VR game (commercial game) | Usual Care | 10 weeks; 6 minutes; 3x week; 180 minutes | Tinetti test Timed Up and Go test Simulator Sickness Questionnaire System Usability Scale |
| Gschwind et al. [36]; Germany, Spain and Australia | Sample: n = 136 participants (VR group = 71; control group = 65) Mean age: VR group = 74.7 ± 6.7; control group = 74.7 ± 6 Gender: VR group = 55.8% female; control group = 66.7% female | iStoppFalls program using Microsoft Kinet with Bumble Bee Park, Hills & Skills, and Balance Bistro games (serious games) | Usual Care | 16 weeks; 60 minutes; 3x week; 2880 minutes | Physiological Profile Assessment European Quality of Life 5 Dimensions 12-item World Health Organization Disability Assessment Schedule 9-item Patient Health Questionnaire Falls Efficacy Scale Incidental and Planned Activity Questionnaire–Spain and Australia Physical Activity Questionnaire–Germany Short Physical Performance Battery Timed Up and Go test Steady-state walking speed—10 m distance Balance test (bipedal, semi-tandem, near tandem, and tandem stance) Sit-to-stand (5 Times) Trail Making Test Victoria Stroop Test Digit Symbol Coding Test Digit Span Backward System Usability Scale 8-item Physical Activity Enjoyment Scale Dynamic Acceptance Model for the Reevaluation of Technologies |
| Schwenk et al. [35]; The EUA | Sample: n = 30 participants (VR group = 15; control group = 15) Mean age: VR group = 84.3 ± 7.3; control group = 84.9 ± 6.6 Gender: VR group = 55.8% female; control group = 68.8% female | Interactive balance training program with 5 wearable inertial sensors (serious game) | No intervention | 4 weeks; 45 minuts; 2x week; 360 minuts | CoM sway area cm2 (BalanSens™) CoM sway area (BalanSens™) Anterior-posterior and medial-lateral (BalanSens™) CoM sway (BalanSens™) Hip sway (deg2) and ankle sway (deg2) (BalanSens™) Reciprocal Compensatory Index (RCI) -Postural coordination strategy (reduction in CoM sway through coordination of hip and ankle motion) Alternate step test Gait Performance (LegSys™) Timed Up and Go test User experience: standardized questionnaire |

IVR, Immersive Virtual reality; VR, Virtual Reality; CoM, Center of Mass.

with CoM sway area (cm$^2$) parameters, antero-posterior CoM sway, and medio-lateral CoM sway [35].

The System Usability Scale was the main instrument used in the studies to assess the usability [34, 36, 37]. One study assessed usability by the user experience using a standardized questionnaire originally developed for evaluating the Wii balance board [35].

In the secondary results, it was observed that two studies tested mobility [34, 36], two studies reported satisfaction [34, 36], two studies measured falls and the risk of falling [35, 36], and two studies evaluated the experience with the game [34, 36]. All studies described that no major adverse events were related to the interventions, and two studies assessed cybersickness using the Simulator Sickness Questionnaire [34, 37].

## Effects on balance and usability

The effects of exergames on functional balance, as measured by the TUG (seconds), were reported in four studies. Campo-Prieto, Cancela-Carral, and Rodríguez-Fuentes [34] found that the control group showed significantly lower performance compared exergame group in TUG test; Campo-Prieto et al. [37] showed that exergame group maintained the total times for the TUG test (−0.45%), and control group had a lower performance; Schwenk et al. [35] demonstrated notably improved performance in the TUG test within the exergame group (effect size = 0.174; P = 0.024); and Gschwind et al. [36] did not observed differences between-group (P = 0.504). The Fig 2A represents the effects of exergaming in the TUG test, post sensitivity analysis. Data suggested that there was a statistically significant difference in the functional balance between the groups, with an effect in favor of the exergame group for the TUG test (MD = -5.90; 95%CI = -10.29 to -1.51; I$^2$ = 25%; low-certainty evidence).

Two studies reported the effects of exergames on the Tinetti test. Campo-Prieto, Cancela-Carral and Rodríguez-Fuentes [34] reported that exergame group showed a significant improvement in Tinetti score (1.84 ± 1.06; p < 0.001); and Campo-Prieto et al. [37] showed that there were statistically significant differences between the groups in the Tinetti test scores for balance (P = 0.004) and total score (P = 0.032), with better performance for the exergame group (P = 0.014). Fig 2B shows the pooled effects of exergames on Tinetti scale indicated an effect in favor of the experimental group (MD = 4.80; 95%CI = 3.36 to 6.24; I$^2$ = 0%; very low-certainty evidence).

In the study of Gschwind et al. [36], There was no difference between groups with respect to the semi-tandem stance, bipedal, and near-tandem stance. Schwenk et al. [35] showed a significant balance improvements in CoM sway area for both eyes closed (P = 0.042; effect size = 0.144) and eyes open (P = 0.007; effect size = 0.239); antero-posterior sway for eyes open (P = 0.015; effect size = 0.201); and medio-lateral sway for eyes open (P = 0.016; effect size = 0.196) and eyes closed (P = 0.012; effect size = 0.214).

With regard to usability outcome, three studies used System Usability Scale (SUS), while one study used the User Experience Questionnaire. The SUS scale provides a simple subjective assessment of the usability (effectiveness, efficiency, and satisfaction) of various products, services, software, hardware, websites, and interface applications [38, 39]. Comprising 10 items, the SUS scale employs a five-point Likert scale (ranging from 1 = totally disagree to 5 = totally agree), resulting in a satisfaction index that ranges from 0 to 100. A higher score indicates better usability of the system. Although the SUS scale is intuitive in many aspects and allows for relative judgments, interpreting the total score in terms of absolute usability remains unclear. In this context, Bangor et al [40] introduced a Likert scale based on this score, which demonstrates a high correlation with the overall SUS score. This enables the classification of systems based on their scores, ranging from 'worst imaginable'

A)

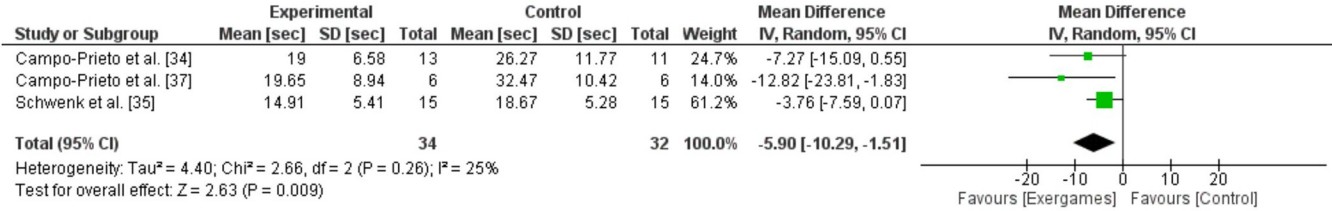

B)

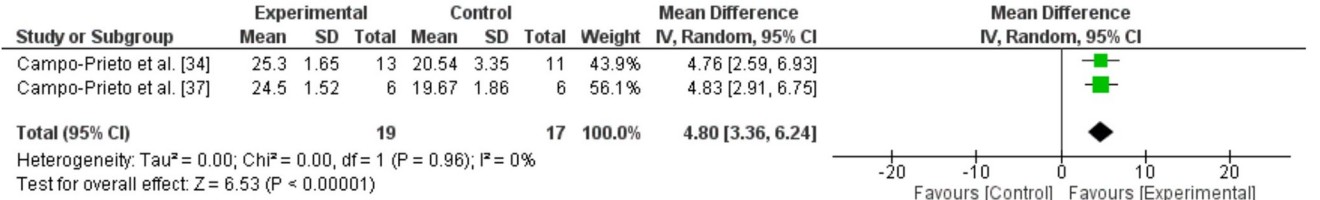

C) (a)

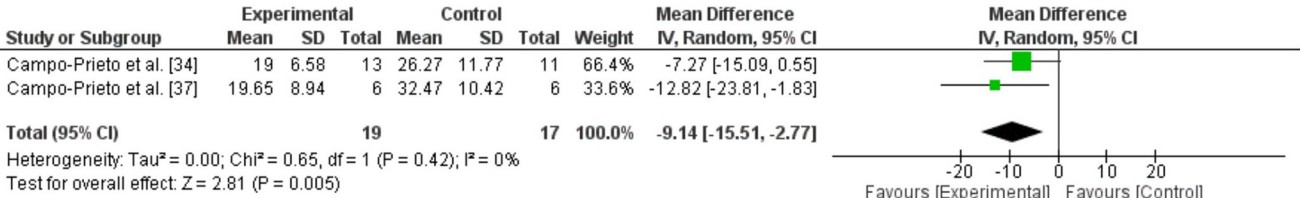

(b)

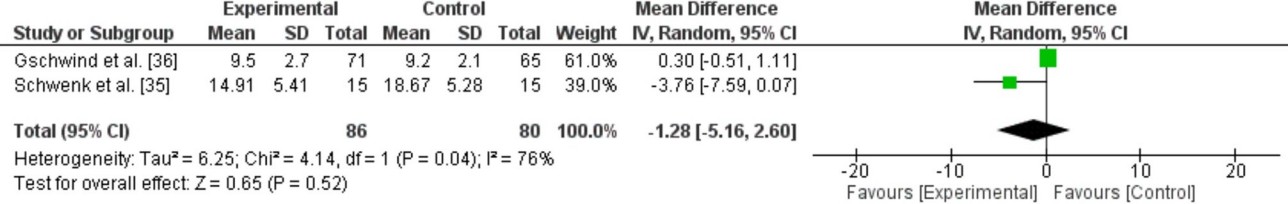

**Fig 2. Forest plot: Effects of exergame interventions home-based in comparison to control group on balance outcome.** (A) Timed Up and Go (TUG) test; (B) Tinetti test; (C) Subgroup analysis with TUG outcome: (a) Immersive exergames, (b) Semi-immersive exergames.

(up to 20.5), 'poor' (21 to 38.5), 'average' (39 to 52.5), 'good' (53 to 73.5), 'excellent' (74 to 85.5), and 'best imaginable' (86 to 100).

The user experience questionnaire allowed the assessment of user experience, utilizing a standardized questionnaire originally developed for evaluating the Wii balance board [35]. This is a 10-question instrument containing responses on a 5-level Likert scale (0 = completely disagree to 4 = absolutely agree, 2 = neutral). The questions are as follows: Q1. It was fun to use the sensor-based balance exercise technology; Q2. Usage of the technology was possible without problems at any time; Q3. I never lost my balance while using the exercise technology; Q4. The form and design of the technology are optimal for me; Q5. I was afraid to tumble or to fall during the exercise; Q6. I required balance support while conducting the exercises; Q7. Thanks to the sensor-feedback, I could quickly learn all exercises; Q8. I feel that the exercises were going too fast for me; Q9. Some of the movements were difficult to perform; Q10. I felt safe using the exercise technology.

The overall average score among studies using SUS scale was 71.43 points, ranging from 62 to 78.33 points. The authors classified the usability score of exergame in the home environment as acceptable [36] and good [34, 37]. According to the SUS scale score classification by Bangor et al. [40], the usability scores of the studies can be categorized as good [34, 36] and excellent [37]. Schwenk et al. [35] used the user experience questionnaire, and suggested that the exergame intervention was feasible and met important requirements of a home training program, including safety and fun to use.

## Subgroup analysis

Subgroup analysis was realized by different immersion levels (semi-immersive and immersive exergames) for the TUG outcome (Fig 2C–(a) and (b)). There was observed a significant difference in the experimental group with immersive exergames when compared to the control group (MD = -9.14; 95%CI = -15.51 to -2.77; $I^2$ = 0%; very low-certainty evidence). No heterogeneity was showed for the immersive exergames studies. However, a substantial heterogeneity was observed for the semi-immersive exergames and no differences between groups (MD = 1.28; 95%CI = -5.16 to 2.60; P = 0.04; $I^2$ = 76%; very low-certainty evidence).

## Effects on secondary outcomes

Campo-Prieto, Cancela-Carral and Rodrígues-Fuentes [34] applied the Five times sit-to-stand test and found that the control group showed significantly a lower performance (an increse of 4.38 seconds in post-intervention) compared to the exergame group, which reduced the test time by 1.75 seconds post-intervention. Gschwind et al. [36] observed no distinctions among the groups in the Short Physical Performance Battery. In the context of an Alternate Step test, Schwenk et al. [35] reported an improvement of 19% in the intervention group (P = 0.037; effect size = 0.151). One trial [34] investigated the quality of life and found that both groups maintained or improved their quality of life scores, mainly the mental score, and the experimental group obtained significantly improved scores in the physical component (P = 0.019) as compared to the control group.

One trial [36] estimated individual fall risk based using the Physiological Profile Assessment and found significantly reduced fall risk in the intervention group compared with the control group (P = 0.035); and the same study did not find significant changes between the groups with the Falls Efficacy Scale. The satisfaction and enjoyment were assessed in two studies. Gschwind et al. [36] reported a mean score of 31 (standard deviation = 8) suggested higher levels of enjoyment with exergame intervention; and Campo-Prieto, Cancela-Carral and Rodrígues-Fuentes [34] found, in the satisfaction questionnaire, a good or very good experiences (100%). The post game experience was assessed by two studies: Campo-Prieto, Cancela-Carral and Rodrígues-Fuentes [34] showed low scores for negative experiences and high for positive experiences, and Gschwind et al. [36] found that the exergame intervention were the most highly rated in terms of appeal, consistency, operation, speed, language and usability. None of the studies examined the motivation of the older adults players in relation to the implementation of the exergame intervention.

## Adverse effects

There were no adverse events reported related to undertaking the interventions in all studies [34–37]. Two studies used the Simulator Sickness Questionnaire and found no symptoms of cybersickness during and after the interventions [34, 37].

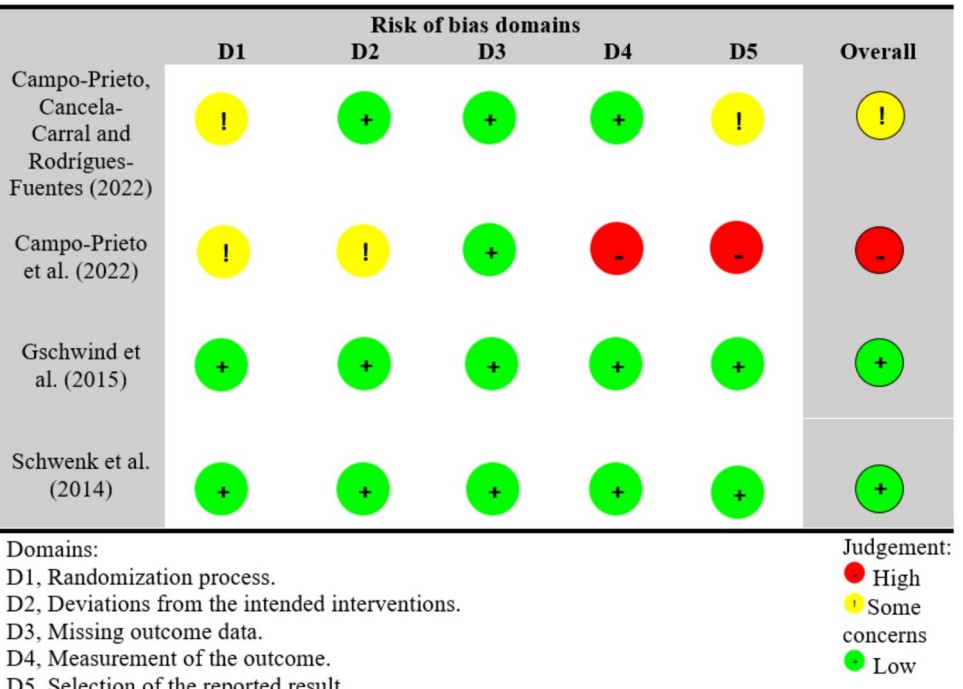

**Fig 3. Methodological quality of the included studies by using the Cochrane Risk-of-Bias tool for randomized trials version 2.**

## Risk of bias and quality of evidence

Fig 3 shows the methodological quality of the included studies. For the randomization process, 50% of the studies showed some concerns [34, 37] and 50% low risk [35, 36]. Regarding deviations from intended interventions, 3 studies showed low risk [34–36]. For the item missing outcome data, all studies showed low risk. Regarding measurement of the outcome, three studies showed low risk [34–36], and one study showed high risk [37]. Finally, two studies showed low risk for selection of the reported result [35, 36], one study reported some concerns [34], and one study showed high risk [37]. For overall methodological quality, two studies [35, 36] showed low risk, one study showed some concerns [34], and one study showed high risk [37]. Table 2 reports the quality of evidence using GRADE. Overall, the certainty of the evidence for outcomes was low to very low.

## Discussion

This review summarized the evidence on the usability of exergames as a balance training tool in the home environment for older adults. We identified that the intervention groups showed better balance outcomes compared to control groups that received usual care or no intervention. Furthermore, exergames demonstrated acceptable and good usability. However, overall, the certainty of the evidence for outcomes was low to very low.

Static and dynamic imbalances are common characteristics of aging and can be effectively addressed through postural and functional balance training, including activities such as reactive recovery techniques and time-reaction exercises [4, 11]. These modalities have the potential to prevent, maintain, or even restore balance in older adults individuals, reducing falls rates [4, 41]. The beneficial effects of exergames on functional balance measures in experimental groups may be attributed to the properties of virtual environment, such as interaction,

**Table 2.** Quality and certainty of evidence of included studies through the Grading of Recommendations Assessment, Development and Evaluation (GRADE) framework.

**Exergames compared to usual care or no intervention for older adults**

**Patient or population:** older adults

**Setting:** home-based environment

**Intervention:** Exergames

**Comparison:** usual care or no intervention

| Outcomes | Anticipated absolute effects[*] (95% CI) | | Relative effect (95% CI) | of participants (studies) | Certainty of the evidence (GRADE) | Comments |
|---|---|---|---|---|---|---|
| | Risk with usual care or no intervention | Risk with Exergames | | | | |
| **Functional balance— Timed Up and Go Test (TUG)** assessed by: seconds | The mean functional balance—TUG test was 0 | MD 5.9 lower (10.29 lower to 1.51 lower) | - | 66 (3 RCTs) | ⊕⊕◯◯◯ Low[a,b] | Exergames may result in a slight increase in functional balance measured by the TUG test. |
| **Tinetti Balance Test** | The mean tinetti balance was 0 | MD 4.8 higher (3.36 higher to 6.24 higher) | - | 36 (2 RCTs) | ⊕◯◯◯ Very low[a,b,c] | Exergames may increase to effect on balance measured by Tinetti test but the evidence is very uncertain. |
| **Functional balance; immersive exergames (TUG)** assessed by: seconds | The mean functional balance; immersive exergames was 0 | MD 9.14 lower (15.51 lower to 2.77 lower) | - | 36 (2 RCTs) | ⊕◯◯◯ Very low[a,b,c] | Immersive exergames may increase to effect on balance measured by TUG test but the evidence is very uncertain. |
| **Functional balance; semi-immersive exergames (TUG)** assessed by: seconds | The mean functional balance; semi-immersive exergames was 0 | MD 1.28 lower (5.16 lower to 2.6 higher) | - | 166 (2 RCTs) | ⊕◯◯◯ Very low[b,d] | The evidence is very uncertain about the effect of semi-immersive exergames. |
| Usability | The mean usability was 0 | 0 (0 to 0) | - | 202 (4 RCTs) | - | Trials could not be pooled due to subjectivity of the outcome assessment. |

*The risk in the intervention group (and its 95% confidence interval) is based on the assumed risk in the comparison group and the relative effect of the intervention (and its 95% CI).

CI: confidence interval; MD: mean difference

**GRADE Working Group grades of evidence**

**High certainty:** we are very confident that the true effect lies close to that of the estimate of the effect.

**Moderate certainty:** we are moderately confident in the effect estimate: the true effect is likely to be close to the estimate of the effect, but there is a possibility that it is substantially different.

**Low certainty:** our confidence in the effect estimate is limited: the true effect may be substantially different from the estimate of the effect.

**Very low certainty:** we have very little confidence in the effect estimate: the true effect is likely to be substantially different from the estimate of effect.

Explanations

a. Downgraded one level due to several ratings with 'unclear' or even 'high' risk of bias.

b. Downgraded one level due to small total population size (< 400) or downgraded two levels due to small total population size (<400) and imprecision of estimation.

c. Downgraded one or two levels due to weight of studies (> 50%).

d. Downgraded one level due to moderate or high heterogeneity (> 50%).

enjoyment, motivation, flow, playfulness, abstraction from reality, challenge, immediate feedback, and engagement, in addition to specific characteristics of the games used. Overall, the games induced instabilities, involving variations in the center of gravity, steps in different directions, transfers and shifts in the participant's weight. Moreover, they work on the general mobility of the body, coordinated movement of the upper and lower limbs, quick reactions involving the trunk and lower limbs, and some cognitive tasks targeting semantic and working memory. Consistent with our study, a meta-analysis found significant effect in favor of exergames regarding TUG test (MD = - 2.48s, 95%CI = - 3.83 to—1.12s) [11]. Taylor et al. [42] showed significant differences in favor of exergames over conventional exercise (MD = 4.33, 95%CI = 2.93 to 5.73) and no intervention (MD = 0.73, 95%CI = 0.17 TO 1.29) for Berg Balance Measure. The meta-analysis conducted by Chen et al. [43] investigated the impact of VR

exergame interventions among older adults living in long-term care facilities. The findings revealed that exergames had a positive effect and could improve the balance ability.

Regarding the subgroup analysis, a significant effect in favor of intervention group compared to usual care was observed for an immersive environment, although with very low-certainty evidence. The balance training in the immersive component may have been the differentiating factor for these results, because the immersive experience is a multisensory modality and provides greater focus of patient attention and task concentration, furthermore, the methodological similarity in the execution of studies that utilized immersive exergames may have influenced these findings. Immersive VR is a term used for technologies that give users a first-person viewpoint, allowing them to engage with virtual worlds in a more realistic way [44]. Utilizing multisensory approaches [45], these systems aim to elevate the level of immersion in the experience with the game execution. The creation of a visual illusion of depth from two images via binocular vision enhances immersion and presence, especially when the visual field motion aligns accurately with head movement [44]. Immersive environments can lead to advantages related to improvements in functional balance, walking speed, and greater overall functionality, and shows promise as an additional resource within the realms of rehabilitation, healthcare, and promoting active aging [44, 46, 47].

A high and substantial heterogeneity was observed in the semi-immersive analyses without difference between groups. A discrepancy among the studies, mainly due to sample size and intervention duration, may have influenced these results for the semi-immersive environment. Hoeg et al. [48] reviewed the immersion system in VR-based rehabilitation of motor function in older adults and pointed out that most clinical interventions utilize semi-immersive systems, ranging from commercial products like Nintendo Wii, to bespoke systems that combine tracking devices, software, and displays. The meta-analysis conducted by Yu et al. [49] showed that semi-immersive VR was more effective in improving cognitive flexibility compared to the other two types of VR (full and non-immersive) for older adults with mild cognitive impairment (MD = - 91.95, 95%CI = - 113.58 to—70.32). Another meta-analysis revealed that non-immersive subgroup analysis for TUG score showed a significant treatment effect on the experimental group [48]. A more robust conclusion that immersive or semi-immersive exergames are or not most effective in clinical practice in the home environment cannot be made due to the lack of experimental studies directly comparing the types of immersion.

Regarding the type of virtual games used, half of the studies used serious games and the other half used commercial games. The use of commercial exergames in clinical settings occurred adaptively to meet the demands of rehabilitation, however, in recent years, a wide range of games has been developed with therapeutic goals and objectives [50]. These games, known as serious games, have a purpose beyond entertainment, allowing for tailored practice to meet the user's needs, and offer immersion, concentration, interaction, targeted feedback, and active participation during rehabilitation [14]. Although there is a substantial body of literature on exergames, there is still a lack of comprehensive information regarding the dose-response relationship. The dose and intensity of the exergames interventions may have been insufficient because they varied greatly across studies in this review. We observed a wide range of exposure time to exergames, ranging from 180 to 2880 minutes–calculated by multiplying the number of sessions by the duration of each session–, and most interventions occurred three times a week ranging from 4 to 16 weeks. Pacheco et al. [11] found an exposure time varied from 360 to 2880 minutes. Miller et al. [25] reported that VR sessions at home lasted from 20 to 75 minutes occurring one to five times per week, for durations of 10 days to 3 months. Chen et al. [43] observed that exergames intervention period ranged from 3 to 15 weeks, with a frequency of 2–3 times per week, and lasted from 18 min to 60 min.

Understanding the usability of exergames and the attitudes of end-users is crucial for the successful use of technology. The success does not solely depend on its effectiveness when used, but also on its likelihood of being utilized in clinical practice of rehabilitation [18]. Older adults individuals frequently possess limited knowledge of technologies. Therefore, it is vital to ensure technology-based training systems that instills technical confidence and guarantees safety; and this can be achieved through features like a straightforward setup, stable connections, and an intuitive gaming environment [41, 51]. Previous studies have highlighted the importance of age-appropriate design and impeccable technical functionality for the usability of exergames [39, 52]. Older adults enjoyed the exergames assessed in this review, and their usability was found to be acceptable and good, making them a positive option for promoting the regular physical activity among older adults in home environments. Furthermore, it stands out that despite the limitations that some older adults might have in operating exergames, the findings of this review propel this field of rehabilitation. The application of exergames is a trend, mainly to aggregate the use of new technologies with active and healthy aging; for being a tool that generates greater interest, attention, curiosity, and pleasure compared to conventional practices; and for the richness of simultaneous motor and cognitive stimuli, which enhance the functional capacity of older adults.

It is important to highlight that none of the studies assessed the motivation. This fact seems paradoxical, since motivation is often a central principle in the reasoning for using the technologies like the modalities of exergame in clinical population [48]. Motivation-related factors appear to affect the outcomes of the exergame intervention in relation to balance performance, and, in particular, the components related to motivation (such as feedback provision) and components associated with capabilities (such as personalized exercises) seem to exert an influence on the overall effectiveness of exergame training [53]. Therefore, it is crucial to take motivational factors into account during exergame interventions for older adults. Despite studies reporting the presence of cybersickness as a side effect [54, 55], particularly in interventions that utilized the immersive modality, no adverse effects were observed in this review, including through the specific assessment of cybersickness. Some adverse events might be generally poorly reported in literature, since while the absence of adverse events could be attributed to their lack of occurrence, it could also be a result of only considering serious events and disregarding negligible, minor symptoms or subtle effects (an example could be how a slight dizziness could easily go unreported) [48].

Some limitations were observed in this systematic review. The studies involving applications of exergames technologies in a home environment were limited, as well as usability research in this context. As the usability measure was assessed only in the exergames group, after the intervention, this prevented more specific analyses from being conducted. The specificity of the PICO elements in this study may have limited the search for scientific studies, despite the utilization of a comprehensive search strategy.

## Conclusion

The effects of exergames were expressive and significant and clinically for functional balance compared to usual care or no intervention, particularly in the immersive modality. The usability of exergames applied in the home environment was considered acceptable and good. However, the certainty of the evidence in this review is low or very low. Therefore, our confidence in the estimated effect was greatly restricted and is expected to change with the conduct of future research. Future studies are required to enhance understanding of the effects of exergaming in a home environment, mainly measures related with motivation, quality of life, and functionality.

## Supporting information

**S1 File. PRISMA checklist.**
(DOCX)

**S2 File. Search strategies for all databases.**
(DOCX)

**S3 File. List of articles.**
(XLSX)

**S4 File. Data.**
(XLSX)

## Acknowledgments

The authors are thankful to the Physical Therapy Postgraduate Program of Federal University of Rio Grande do Norte, Natal, Brazil, for providing support the study.

## Author Contributions

**Conceptualization:** Candice Simões Pimenta de Medeiros, Rummenigge Rudson Dantas, Fabrícia Azevedo da Costa Cavalcanti.

**Data curation:** Candice Simões Pimenta de Medeiros, Fabrícia Azevedo da Costa Cavalcanti.

**Formal analysis:** Candice Simões Pimenta de Medeiros.

**Methodology:** Candice Simões Pimenta de Medeiros, Luanna Bárbara Araújo Farias.

**Project administration:** Maria Clara do Lago Santana.

**Writing – original draft:** Candice Simões Pimenta de Medeiros, Luanna Bárbara Araújo Farias, Thaiana Barbosa Ferreira Pacheco, Rummenigge Rudson Dantas, Fabrícia Azevedo da Costa Cavalcanti.

**Writing – review & editing:** Candice Simões Pimenta de Medeiros, Luanna Bárbara Araújo Farias, Maria Clara do Lago Santana, Thaiana Barbosa Ferreira Pacheco, Rummenigge Rudson Dantas, Fabrícia Azevedo da Costa Cavalcanti.

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
