## [Decision Letter · Decision Letter 0]

14 Feb 2024

PONE-D-23-27194A systematic review of exergame usability as home-based balance training tool for older adultsPLOS ONE

Dear Dr. Medeiros,

Thank you for submitting your manuscript to PLOS ONE. After careful consideration, we feel that it has merit but does not fully meet PLOS ONE’s publication criteria as it currently stands. Therefore, we invite you to submit a revised version of the manuscript that addresses the points raised during the review process.

We look forward to receiving your revised manuscript.

Kind regards,

Esedullah Akaras

Academic Editor

PLOS ONE

Journal Requirements:

Comparison of the effects of virtual reality-based balance exercises and conventional exercises on balance and fall risk in older adults living in nursing homes in Turkey - https://doi.org/10.3109/09593985.2015.1138009

In your revision ensure you cite all your sources (including your own works), and quote or rephrase any duplicated text outside the methods section. Further consideration is dependent on these concerns being addressed.

Reviewers' comments:

Reviewer's Responses to Questions

**Comments to the Author**

1. Is the manuscript technically sound, and do the data support the conclusions?

Reviewer #1: Yes

Reviewer #2: Partly

2. Has the statistical analysis been performed appropriately and rigorously? 

Reviewer #1: I Don't Know

Reviewer #2: Yes

3. Have the authors made all data underlying the findings in their manuscript fully available?

Reviewer #1: Yes

Reviewer #2: No

4. Is the manuscript presented in an intelligible fashion and written in standard English?

Reviewer #1: Yes

Reviewer #2: No

5. Review Comments to the Author

Reviewer #1: Firstly, this review has been conducted well in the aspect of both methodology and significance that needs researchers to weigh. Although I have some criticisms about the modalities of exergames and balance assessments, the authors are also aware of these issues and express them in the discussion section of this article. Hence, this review has a significant potential to highlight the efficacy of this modality in pre and post-rehabilitation in every area of physiotherapy and also to enlighten future research.

Reviewer #2: 1) Technically sound, do the data support the conclusions?

This study sought to review the current literature on the usability of exergames in older adults’ home environment and its effect on balance outcomes.

The manuscript appears technically sound in that it follows systematic review and meta analysis guidelines established by the Cochrane Collaboration and others in regards to the Timed Up and Go outcome. There were two major concerns that limit the ability to support the conclusions. These are 1) the operational definition of a person’s home and why it is important. Is it to increase independent use? Improve autonomy? Or are older adults expected to have a caregiver support in order to use exergaming systems? This distinction is important because having an older adult play an on their Oculus at home is very different from the recreation organizer a community based senior center or residential care facility setting the game up for them. So to improve, clarify whether or not the older adults would have help or not as this would greatly influence the usability outcome. The second concern is the measures of “usability” are not described clearly enough to convince the reader what the “acceptable” “good” and “feasible” tool actually mean. The term itself is defined, but not the scale or perhaps other means of measuring the subdomains of “usability”. Or, in the intro, why you are using the TUG is to capture the effectiveness domain.

2) Analysis: yes, also using established methods

3) No. Usability scale data is not clearly available

4) In regards to overall sentence and paragraph structure, it is intelligible with the exception of an occasional questionable word choice (“worsened” in the results section). The more challenging issue is conceptual and operational definitions described above (home, how usability is measured) and the universal challenge of describing what “balance” or “postural control” mean. Suggestion: establish terms in the beginning, whether it is static / dynamic, functional mobility (gait and sit to stand, which will be TUG and the Tinetti POMA) and be consistent with their use throughout the entire manuscript.

Details

Page 3

paragraph 2: ref 4- “several studies”, “most populations”- seems much too broad and non specific. Also, did this reference have data on fun, attractive, viable or was this a discussion item? Are there data in this paper to support this claim?

Paragraph 3 Usability as defined as efficiency, effectiveness, and satisfaction. This operational definition is important, and a reasonable start. One of the major concerns this the methodology of the paper is that is does not describe the questionnaires designed to measure usability sufficiently to allow the reader to interpret the results.

Pg 4

“Impact on postural control, mobility, QOL, adverse effects” is QOL measured? How?

What is considered a “home environment”? Senior citizens center might mean different things to different people. I wonder about “residential care environment” and autonomy

Pg 5

Outcomes: listed as 1) “postural balance” and 2) usability. For consistency, keep the order the same throughout the manuscript- see the results section in which they are reversed. As for the terminology, often “postural control”, “balance (static vs dynamic)”, or “functional balance” are used interchangeably. Keep consistent and recognize that these outcomes measure many different resources for postural control. Using systems theory (Horak et al 2006, I believe, the basis for the miniBESTest) might be a helpful conceptual anchor.

Secondary results: What does “mobility “ mean? Some could consider the timed up and go “ functional mobility”

Results:

Flow of studies: Generally acceptable. Check writing guidelines if they prefer starting the sentence with the number written out or if numerals are acceptable.

Characteristics:

I have concerns that a “senior center” is conducted in the community, with staff support, as is the “senior living community”

Usability is listed first, then balance

Scores: can these not be used for a meta analysis? What is the usability score – what does it mean?

Pg 10

Effects on Usability and Balance

Usability is also satisfactorily defined in the introduction, but the main outcome – System Usability Scale was not described enough to make any interpretations. How many points is it out of? Does it ask questions encompassing the three domains of usability? If it is too subjective to use in a meta-analysis, why are the data reported to the hundredths place?

Exergames on “functional balance” – new category? Previously all were lumped together as “postural balance”

Discussing reference (25) – “the control group significantly worsened compared to the exergame group” The use of “worsened” is completely inappropriate here as it means the control groups’s TUG scores increased over time. If there is not pre-post data, just post intervention data, all one can say is that the TUG scores were significantly higher in the control. Further, better or worse should be saved for the discussion as it implies judgement

Effects on secondary outcomes

Again with the worsened.

Why is there a statement about motivation?

Adverse effects : No complaints aside from the extra s in ‘adverses’

Pg 15-

I don’t believe you have to show the actual effect size data from other studies, but tell the readers how these were also different from your research?

Pg 16

Refs 37, 39, 40- what is meant by motor control and functionality? Compared to what?

Pg 17

Why is QOL scores here in the discussion?

18-

Is this paragraph a discussion of your secondary measures or what was not measured? Why picking out motivation when it was not your research question?

6. PLOS authors have the option to publish the peer review history of their article (what does this mean?). If published, this will include your full peer review and any attached files.

Reviewer #1: No

Reviewer #2: No

---

## [Author Response · Author response to Decision Letter 0]

22 Apr 2024

RESPONSE TO REVIEWERS

Response from Medeiros et al.

We are grateful for all comments and suggestions raised by Reviewer 1 and Reviewer 2.

• Journal Requirements

A) Please ensure that your manuscript meets PLOS ONE's style requirements, including those for file naming. The PLOS ONE style templates can be found at 

Response from Medeiros et al.

As recommended, we have ensured that the manuscript conforms to the style requirements.

B) Note from Emily Chenette, Editor in Chief of PLOS ONE, and Iain Hrynaszkiewicz, Director of Open Research Solutions at PLOS: Did you know that depositing data in a repository is associated with up to a 25% citation advantage (https://doi.org/10.1371/journal.pone.0230416)? If you’ve not already done so, consider depositing your raw data in a repository to ensure your work is read, appreciated and cited by the largest possible audience. You’ll also earn an Accessible Data icon on your published paper if you deposit your data in any participating repository (https://plos.org/open-science/open-data/#accessible-data).

Response from Medeiros et al.

The authors would like to thank the Editors for the suggestion. We have included the study data in the supporting information.

C) We noticed you have some minor occurrence of overlapping text with the following previous publication(s), which needs to be addressed:

Comparison of the effects of virtual reality-based balance exercises and conventional exercises on balance and fall risk in older adults living in nursing homes in Turkey - https://doi.org/10.3109/09593985.2015.1138009. In your revision ensure you cite all your sources (including your own works), and quote or rephrase any duplicated text outside the methods section. Further consideration is dependent on these concerns being addressed.

Response from Medeiros et al.

Thank you for bringing this to our attention. We have rectified the overlapping text and utilized two distinct programs to identify any additional potential overlaps. We appreciate your feedback, and we have cited all sources in our revision.

D) Please include captions for your Supporting Information files at the end of your manuscript, and update any in-text citations to match accordingly. Please see our Supporting Information guidelines for more information: http://journals.plos.org/plosone/s/supporting-information. 

Response from Medeiros et al.

We included captions in our supporting information files, and updated the text citations.

• Reviewer #1

A) Firstly, this review has been conducted well in the aspect of both methodology and significance that needs researchers to weigh. Although I have some criticisms about the modalities of exergames and balance assessments, the authors are also aware of these issues and express them in the discussion section of this article. Hence, this review has a significant potential to highlight the efficacy of this modality in pre and post-rehabilitation in every area of physiotherapy and also to enlighten future research.

Response from Medeiros et al.

The authors would like to thank the reviewer for the comprehensive comments. We sincerely appreciate your acknowledgment of the methodology and significance of our research. Your feedback is invaluable to us.

• Reviewer #2

A) 1) Technically sound, do the data support the conclusions? This study sought to review the current literature on the usability of exergames in older adults’ home environment and its effect on balance outcomes.

The manuscript appears technically sound in that it follows systematic review and meta analysis guidelines established by the Cochrane Collaboration and others in regards to the Timed Up and Go outcome. 

There were two major concerns that limit the ability to support the conclusions. These are: 

1) the operational definition of a person’s home and why it is important. Is it to increase independent use? Improve autonomy? Or are older adults expected to have a caregiver support in order to use exergaming systems? This distinction is important because having an older adult play an on their Oculus at home is very different from the recreation organizer a community based senior center or residential care facility setting the game up for them. So to improve, clarify whether or not the older adults would have help or not as this would greatly influence the usability outcome. The second concern is the measures of “usability” are not described clearly enough to convince the reader what the “acceptable” “good” and “feasible” tool actually mean. The term itself is defined, but not the scale or perhaps other means of measuring the subdomains of “usability”. Or, in the intro, why you are using the TUG is to capture the effectiveness domain.

2) Analysis: yes, also using established methods

3) No. Usability scale data is not clearly available

4) In regards to overall sentence and paragraph structure, it is intelligible with the exception of an occasional questionable word choice (“worsened” in the results section). The more challenging issue is conceptual and operational definitions described above (home, how usability is measured) and the universal challenge of describing what “balance” or “postural control” mean. Suggestion: establish terms in the beginning, whether it is static / dynamic, functional mobility (gait and sit to stand, which will be TUG and the Tinetti POMA) and be consistent with their use throughout the entire manuscript.

Response from Medeiros et al.

The authors would like to thank you for providing such detailed comments, which are very constructive and will help to improve the manuscript. We appreciate your acknowledgment of the technical soundness of our manuscript. We strived to address all the reviewer questions and requests.

The home environment is a dynamic setting characterized by comfort, security, and privacy, where people express their personal identity, experience autonomy, and which influences the level of physical activity that older adults perform during their daily routines (Meghani NAA, Hudson J, Stratton G, Mullins J. Older adults' perspectives on physical activity and sedentary behaviour within their home using socio-ecological model. Plos One, 2023, 20;18(11):e0294715; Aclan R, George S, Block H, Lane R, Laver K. Middle age and older adult’s perspectives of their own home environment: a review of qualitative studies and meta-synthesis. BMC Geriatric, 2023, 31;23(1):707). Therefore, practicing physical exercise at home is important to attenuate the risk of falls, improve aerobic fitness, and reduce sedentary behavior in this population, enhancing functional capacity and autonomy. We added a summary of this description in the introduction of the manuscript.

As suggested, we clarified whether or not the older adults had assistance in preparing, using, and supervising the use of exergames. This information was added in the results section under the topic of characteristics of participants and interventions.

As suggested, the usability measures were described more clearly, including instruments used and their means of measuring the subdomains. Additionally, data from the usability scales were made available.

The Timed Up and Go Test (TUG) was used in this study as one of the tools to assess functional balance. Furthermore, the results were analyzed using the TUG, as it was the main instrument used among the studies included in this review. The TUG is considered a reliable tool for assessing general balance, functional mobility, risk of falls, and its shows a significant correlation with fear of falling and functional performance (Ortega-Bastidas P, Gómez B, Aqueveque P, Luarte-Martínez S, Cano-de-la-Cuerda R. Instrumented Timed Up and Go Test (iTUG)—More Than Assessing Time to Predict Falls: A Systematic Review, Sensors, 2023; 23(7):3426; Barry E, Galvin R, Keogh C, Horgan F, Fahey T. Is the Timed Up and Go test a useful predictor of risk of falls in community dwelling older adults: a systematic review and meta- analysis. BCM Geriatrics, 2014; 14:14; Panel on Prevention of Falls in Older Persons. American Geriatrics Society and British Geriatrics Society. Summary of the Updated American Geriatrics Society/British Geriatrics Society clinical practice guideline for prevention of falls in older persons. J Am Geriatr Soc. 2011;14(1):148–157). Additionally, it is recommend for assessing gait and balance to prevent falls in older adults (NICE. The assessment and prevention of falls in older people. 2013).

We have adjusted questionable words throughout the text. The conceptual definitions of ‘home’ and ‘usability’ were clarified in the manuscript. We defined postural control, as well as functional balance, in the introduction of the article. Furthermore, we standardized the use of the term ‘functional balance’ throughout the article.

B) Details: Page 3 – Paragraph 2: ref 4- “several studies”, “most populations”- seems much too broad and non specific. Also, did this reference have data on fun, attractive, viable or was this a discussion item? Are there data in this paper to support this claim?; Paragraph 3 Usability as defined as efficiency, effectiveness, and satisfaction. This operational definition is important, and a reasonable start. One of the major concerns this the methodology of the paper is that is does not describe the questionnaires designed to measure usability sufficiently to allow the reader to interpret the results.

Response from Medeiros et al.

Thank you for this suggestion. In paragraph 2, we further specify the reference 4. While this reference supports the information, we have included additional references to reinforce this point.

We appreciate the feedback on paragraph 3. We have decided to describe the usability evaluation questionnaires in the results section – specifically, under the topic of effects on balance and usability – in order to allow the reader to better interpret the results.

C)Details: Pg 4 “Impact on postural control, mobility, QOL, adverse effects” is QOL measured? How? 

What is considered a “home environment”? Senior citizens center might mean different things to different people. I wonder about “residential care environment” and autonomy

Response from Medeiros et al.

We considered analyzing aspects related to quality of life and motivation in the study protocol and throughout the analysis of this review. However, upon reviewing the data, we found that only one study investigated quality of life, and no study analyzed participants' motivation. Therefore, we included this information in the result section under the effects on secondary outcomes.

Thank you for your consideration. We appreciate your suggestion, and upon review, we acknowledge that we did not clarify this information adequately. Therefore, we have adjusted our text to emphasize that the inclusion criteria for the study require older adults to exhibit autonomy and significant functional capacity. This clarification is essential due to the varying interpretations of the senior citizens’ clubs, elderly homes or retirement homes, residential care facilities, assisted living communities, and independent living centers, which may differ among individuals and location worldwide.

D) Details: Pg 5 Outcomes: listed as 1) “postural balance” and 2) usability. For consistency, keep the order the same throughout the manuscript- see the results section in which they are reversed. As for the terminology, often “postural control”, “balance (static vs dynamic)”, or “functional balance” are used interchangeably. Keep consistent and recognize that these outcomes measure many different resources for postural control. Using systems theory (Horak et al 2006, I believe, the basis for the miniBESTest) might be a helpful conceptual anchor.

Secondary results: What does “mobility “ mean? Some could consider the timed up and go “ functional mobility”

Response from Medeiros et al.

As suggested, we have adjusted the order of information regarding balance and usability throughout the text. We adopted the concepts from the study by Horak et al. (2006), and we recognize that these results measure many different aspects of postural balance in the outcomes measures section.

Mobility is a broad term, defined as the ability to move around and change positions, such as walking, rising from a chair, and maintaining balance while standing. Thus, mobility comprises all the skills required for everyday living: physical resistence, strength, balance, coordination, and range of motion (Treacy D, Hassett L, Schurr K, Fairhall NJ, Cameron ID, Sherrington C. Mobility training for increasing mobility and functioning in older people with frailty. Cochrane Database Sys Rev. 2022 30;6(6): CD010494).

The TUG test has been extensively researched and widely used in clinical environments to assess balance and mobility for over 20 years. It quantifies several different elements of mobility and has been frequently employed in assessing fall risk. However, some studies consider the TUG test as part of functional balance analysis (Agathos CP, Velisar A, Shanidze N. A Comparison of Walking Behavior during the Instrumented TUG and Habitual Gait. Sensors, 2023, 18;23(16):7261; Sedaghati P, Goudarzian M, Ahmadabadi S, Tabatabai-Asl S. The impact of a multicomponent-functional training with postural correction on functional balance in the elderly with a history of falling. J Exp Orthop, 2022, 9:23; Jung J, Kim MG, Kang YJ, Min K, Han KA, Choi H. Vibration Perception Threshold and Related Factors for Balance Assessment in Patients with Type 2 Diabetes Mellitus. Int J Environ Res Public Health, 2021, 4;18(11):6046; Yu L, Zhao Y, Wang H, Sun TL, Murphy TE, Tsui KL. Assessing elderly’s functional balance and mobility via analyzing data from waist-mounted tri-axial wearable accelerometers in timed up and go tests. BCM Med Inform Decis Mak. 2021, 21:108; Yeşilyaprak SS, Yıldırım M, Tomruk M, Ertekin O, Algun ZC. Comparison of the effects of virtual reality-based balance exercises andconventional exercises on balance and fall risk in older adults living in nursinghomes in Turkey. Physiotherapy theory andpractice. 2016, 32(3):191-201). Furthermore, the TUG test has several other applications and has been used, for instance, as a tool for sarcopenia screening, givin its capacity to assess muscular strength and speed in a single test (Queiroz LL, Silva LGO, Pinheiro HA. Can the timed up and go test be used as a predictor of muscle strength in older adults?. Fisiot Pesq, 2023; 30:ee22013723; Filippin LI, Miraglia F, Teixeira VNO, Boniatti MM. Timed Up and Go test as a sarcopenia screening tool in home-dwelling elderly persons. Rev Bras Geriatr Gerontol. 2017;20(4):561-6).

In this sense, considering that the task performed by the TUG test is significant for daily life and the risk of falls, as it includes several important postural transitions (sitting to standing, starting to walk, turning) and movements more susceptible to loss of balance in the older adults, we chose to categorize this instrument as an outcome of postural/functional balance. Furthermore, functional balance is generally assessed using the main instruments selected for analysis in this review, such as the Berg Balance Scale, TUG test, Functional Reach Test, among others. 

E) Results: Flow of studies: Generally acceptable. Check writing guidelines if they prefer starting the sentence with the number written out or if numerals are acceptable.

- Characteristics: I have concerns that a “senior center” is conducted in the community, with staff support, as is the “senior living community”

- Usability is listed first, then balance

- Scores: can these not be used for a meta analysis? What is the usability score – what does it mean?

Pg 10

Response from Medeiros et al.

 Thanks for the comments. We checked the guidelines and aligned our text.

In response to the reviewer's concerns, we would like to clarify that two studies included in our review conducted the intervention at the same senior center, the Saraiva Senior Center of Pontevedra in Spain. After researching the institution, we discovered th

---

## [Decision Letter · Decision Letter 1]

8 May 2024

PONE-D-23-27194R1A systematic review of exergame usability as home-based balance training tool for older adultsPLOS ONE

Dear Dr. Medeiros,

Thank you for submitting your manuscript to PLOS ONE. After careful consideration, we feel that it has merit but does not fully meet PLOS ONE’s publication criteria as it currently stands. Therefore, we invite you to submit a revised version of the manuscript that addresses the points raised during the review process.

Your article will be accepted after the relevant minor corrections are made.

We look forward to receiving your revised manuscript.

Kind regards,

Esedullah Akaras

Academic Editor

PLOS ONE

Journal Requirements:

Reviewers' comments:

Reviewer's Responses to Questions

**Comments to the Author**

1. If the authors have adequately addressed your comments raised in a previous round of review and you feel that this manuscript is now acceptable for publication, you may indicate that here to bypass the “Comments to the Author” section, enter your conflict of interest statement in the “Confidential to Editor” section, and submit your "Accept" recommendation.

Reviewer #1: All comments have been addressed

Reviewer #2: All comments have been addressed

2. Is the manuscript technically sound, and do the data support the conclusions?

Reviewer #1: Yes

Reviewer #2: Yes

3. Has the statistical analysis been performed appropriately and rigorously? 

Reviewer #1: I Don't Know

Reviewer #2: Yes

4. Have the authors made all data underlying the findings in their manuscript fully available?

Reviewer #1: Yes

Reviewer #2: No

5. Is the manuscript presented in an intelligible fashion and written in standard English?

Reviewer #1: Yes

Reviewer #2: Yes

6. Review Comments to the Author

Reviewer #1: (No Response)

Reviewer #2: re- #4. The link to the supporting data did not work- so this might be a problem on my end.

Overall previous edits were addressed effectively. With the intent of helping you make this paper the best it can me, I found two things:

104- A repeat of line 97, and confusing: at first I thought this was an incomplete sentence, details as a noun instead of a verb, and used being the operative verb here.

330-332. sorry if I did not catch this last time, this statement needs clarification:

310. “the Five times sit-to-stand test and found that the control group showed significantly a lower performance compared exergame group that reduced the time of the test in the post intervention. “

In a timed test, a lower score = faster, so this means the control group improved, not the intervention group.

7. PLOS authors have the option to publish the peer review history of their article (what does this mean?). If published, this will include your full peer review and any attached files.

Reviewer #1: **Yes: **Gökhan Mehmet Karatay

Reviewer #2: No

---

## [Author Response · Author response to Decision Letter 1]

21 Jun 2024

RESPONSE TO REVIEWERS

Response from Medeiros et al.

We are grateful for all comments and suggestions raised by Reviewer 2.

• Journal Requirements

Response from Medeiros et al.

As recommended, we have reviewed the reference list to ensure it is correct. Nine articles were added to the manuscript. Most of the studies added were to complement the introduction, and one study was added to add to enhance the description of usability. The articles added to the manuscript are:

(4) Yeşilyaprak SS, Yildirim MŞ, Tomruk M, Ertekin Ö, Algun ZC. Comparison of the effects of virtual reality-based balance exercises and conventional exercises on balance and fall risk in older adults living in nursing homes in Turkey. Physiother Theory Pract. 2016;32(3):191–201.

(5) Horak F. Postural orientation and equilibrium: what do we need to know about neural control of balance to prevent falls? Age Aging. 2006;35(2):7–11.

(6) Sedaghati P, Goudarzian M, Ahmadabadi S, Tabatabai-Asl SM. The impact of a multicomponent-functional training with postural correction on functional balance in the elderly with a history of falling. J Exp Orthop. 2022;9(1).

(7) Alshahrani MS, Reddy RS. Kinesiophobia, limits of stability, and functional balance assessment in geriatric patients with chronic low back pain and osteoporosis: a comprehensive study. Front Neurol. 2024; 15:1354444.

(8) Dunsky A. The Effect of Balance and Coordination Exercises on Quality of Life in Older Adults: A Mini-Review. Front Aging Neurosci. 2019;11(318):1–10.

(10) Rose T, Nam CS, Chen KB. Immersion of virtual reality for rehabilitation - Review. Vol. 69, Applied Ergonomics. Elsevier Ltd; 2018. p. 153–61.

(19) Aclan R, George S, Block H, Lane R, Laver K. Middle aged and older adults’ perspectives of their own home environment: a review of qualitative studies and meta-synthesis. BMC Geriatr. 2023;23(707):1–12.

(20) Meghani NAA, Hudson J, Stratton G, Mullins J. Older adults’ perspectives on physical activity and sedentary behavior within their home using socio-ecological model. PLoS One. 2023;18(11):1–24.

(38) Brooke J. SUS - A quick and dirty usability scale. Usability Eval Ind. 1996;189–94.

• Reviewer #1 

(No Response)

• Reviewer #2

A) re- #4. The link to the supporting data did not work- so this might be a problem on my end.

Response from Medeiros et al.

Thank you for this comment. We will attach all supporting information files individually and in a compressed folder (zipe file).

B) Overall previous edits were addressed effectively. With the intent of helping you make this paper the best it can me, I found two things:

1) 104- A repeat of line 97, and confusing: at first I thought this was an incomplete sentence, details as a noun instead of a verb, and used being the operative verb here.

2) 330-332. sorry if I did not catch this last time, this statement needs clarification: 310. “the Five times sit-to-stand test and found that the control group showed significantly a lower performance compared exergame group that reduced the time of the test in the post intervention. “In a timed test, a lower score = faster, so this means the control group improved, not the intervention group.

Response from Medeiros et al.

The authors would like to thank you for providing such detailed comments, which are very constructive and will help to improve the manuscript. 

1) Thank you for pointing this out. As line 104 was confusing (“The protocol details the complete search strategy used.”), we have chose to remove this excerpt from the manuscript.

2) In fact, we intended to indicate that the reduction in test time post-intervention belonged to the experimental group. We acknowledge that the excerpt highlighted by the reviewer was confusing. 

# Study data (Campo-Prieto P, Cancela-Carral JM, Rodríguez-Fuentes G. Feasibility and Effects of an Immersive Virtual Reality Exergame Program on Physical Functions in Institutionalized Older Adults: A Randomized Clinical Trial. Sensors. 2022;22(6742):1–15):

• Experimental group (seconds) = Pre (15.56±4.52), Post (13.81±3.46), Variation (-1.75). 

• Control group (seconds) = Pre (21.19±12.63), Post (25.57±14.15), Variation (+4.38)* p<0.005. 

• The study by Campo-Prieto, Cancela-Carral and Rodrígues-Fuentes (2022) showed that the control group exhibited a significantly worsening compared to the experimental group.

Therefore, we have adjusted the text and included the performance variation for each group to carify the information.

---

## [Editor Report · Decision Letter 2]

24 Jun 2024

A systematic review of exergame usability as home-based balance training tool for older adults

PONE-D-23-27194R2

Dear Dr. Medeiros,

We’re pleased to inform you that your manuscript has been judged scientifically suitable for publication and will be formally accepted for publication once it meets all outstanding technical requirements.

Kind regards,

Esedullah Akaras

Academic Editor

PLOS ONE
---

## [Editor Report · Acceptance letter]

12 Aug 2024

PONE-D-23-27194R2 

PLOS ONE

Dear Dr. Medeiros, 

I'm pleased to inform you that your manuscript has been deemed suitable for publication in PLOS ONE. Congratulations! Your manuscript is now being handed over to our production team.

Kind regards, 

on behalf of

Dr. Esedullah Akaras 

Academic Editor

PLOS ONE